# The Diagnostic Value of MRI-Based Radiomic Analysis of Lacrimal Glands in Patients with Sjögren’s Syndrome

**DOI:** 10.3390/ijms231710051

**Published:** 2022-09-02

**Authors:** Delia Doris Muntean, Maria Bădărînză, Paul Andrei Ștefan, Manuela Lavinia Lenghel, Georgeta Mihaela Rusu, Csaba Csutak, Paul Alexandru Coroian, Roxana Adelina Lupean, Daniela Fodor

**Affiliations:** 1Radiology Department, Iuliu Hatieganu University of Medicine and Pharmacy, 400012 Cluj-Napoca, Romania; 22nd Internal Medicine Department, Iuliu Hatieganu University of Medicine and Pharmacy, 400006 Cluj-Napoca, Romania; 3Division of Neuro- and Musculoskeletal Radiology, Department of Biomedical Imaging and Image-Guided Therapy, Medical University of Vienna, General Hospital of Vienna (AKH), 1090 Vienna, Austria; 4Anatomy and Embryology, Morphological Sciences Department, Iuliu Hatieganu University of Medicine and Pharmacy, 400012 Cluj-Napoca, Romania; 5Radiology Department, Emergency County Hospital, 400006 Cluj-Napoca, Romania; 6Histology, Morphological Sciences Department, “Iuliu Hațieganu” University of Medicine and Pharmacy, 400012 Cluj-Napoca, Romania

**Keywords:** lacrimal glands, MRI, primary Sjögren’s Syndrome, radiomics, textural analysis

## Abstract

This study aimed to assess the effectiveness of MRI-based texture features of the lacrimal glands (LG) in augmenting the imaging differentiation between primary Sjögren’s Syndrome (pSS) affected LG and healthy LG, as well as to emphasize the possible importance of radiomics in pSS early-imaging diagnosis. The MRI examinations of 23 patients diagnosed with pSS and 23 healthy controls were retrospectively included. Texture features of both LG were extracted from a coronal post-contrast T1-weighted sequence, using a dedicated software. The ability of texture features to discriminate between healthy and pSS lacrimal glands was performed through univariate, multivariate, and receiver operating characteristics analysis. Two quantitative textural analysis features, RunLengthNonUniformityNormalized (RLNonUN) and Maximum2DDiameterColumn (Max2DDC), were independent predictors of pSS-affected glands (*p* < 0.001). Their combined ability was able to identify pSS LG with 91.67% sensitivity and 83.33% specificity. MRI-based texture features have the potential to function as quantitative additional criteria that could increase the diagnostic accuracy of pSS-affected LG.

## 1. Introduction

Primary Sjögren’s Syndrome (pSS) represents a chronic autoimmune inflammatory disease that affects the exocrine glands, especially the salivary (SG) and lacrimal glands (LG), causing xerostomia and xerophthalmia. The pathological changes are due to the periductal lymphocytic infiltration and the consecutive progressive loss of the exocrine glands’ secretory function [1,2]. Patients with pSS have an increased risk (up to 40-fold higher than in the general population) of developing non-Hodgkin’s lymphoma, the main cause of pSS-related mortality [3]. Therefore, it is of paramount importance to choose the optimal imaging technique that allows us not only to diagnose this disease, but also to evaluate its stage and activity in order to start treatment as early as possible [4,5].

The current ACR-EULAR diagnostic criteria for pSS [6] imply that each individual must reach a score greater than 4 when the weights from the five following criteria items are summed: autoantibody positivity anti-Ro/anti-La (3 points), labial salivary gland biopsy revealing focal lymphocytic sialadenitis and a focus score greater ≥ 1 (3 points), at least one eye with an ocular staining score ≥ 5/van Bijsterfeld score ≥ 4 (1 point), at least one eye with a Schirmer test ≤ 5 mm/5 min (1 point), and an unstimulated whole saliva flow rate ≤ 0.1 mL/min (1 point).

In patients with pSS, SG imaging is mainly performed. Ultrasonography of the parotid gland represents the most widely used method in assessing parenchymal changes, clinical activity [7], and the response to therapy [8]. Magnetic resonance imaging (MRI) is also used in evaluating patients with pSS; MRI sialography represents the gold standard imaging technique in staging the disease [9]. The imaging aspect of the parotid and submandibular glands consists of diffuse parenchymal changes, with progressive acinar atrophy and the presence of multiple cystic areas that correspond to the ectasia of the salivary ducts. In advanced stages, the SG parenchyma is completely destructed [8].

LG are paired, almond-shaped glands situated in the superolateral aspect of the orbit, in the extraconal space (Figure 1). They consist of a palpebral and an orbital lobe [10].

As far as LG are concerned, in pSS, both lobes are diffusely affected [10,11]. However, the literature regarding the imaging aspect of the LG in pSS is scarce. One preliminary study proved that the two most relevant ultrasonographic LG features that could help differentiate pSS subjects from healthy subjects were the glandular parenchymal inhomogeneity and the fibrous aspect of LG [12]. One older study proved that MRI was useful in assessing the change in the size of LG in pSS according to the stage of the disease, with four patterns being described: hypertrophic LG, heterogenous normal-sized LG with fat deposition, increased fatty degenerated LG, and atrophic LG [13]. Moreover, on diffusion-weighted imaging (DWI), LG in pSS patients presented lower apparent diffusion coefficient values compared to normal LG of age-matched healthy subjects [14].

Recently, radiology has slowly shifted to radiomics, which has emerged as an innovative method of the quantitative post-processing of medical images through advanced mathematical analysis. The hypotheses underlying radiomics are the following: images represent the phenotypic expression of biological processes, respectively, an image contains much more information than the human eye can perceive [15]. In summary, it is theorized that a pathological process that alters the tissue produces a modified MRI signal, which will, in turn, give textural features different values from those of the normal structures [16,17]. In the last decade, several studies have been published that have analyzed the contributions of radiomics in the fields of head and neck imaging, emphasizing its potential to increase diagnostic accuracy and providing valuable information that could influence and facilitate a therapeutic decision [18,19]. Regarding LG pathology, however, only two studies have been performed so far, and they assessed the applicability of textural analysis on MRI images in differentiating between benign and malignant lesions of LG. Lecler et al. [20] showed that MRI-based texture features could act as biomarkers for the detection of LG tumors, while Guo et al. [21] demonstrated that radiomic features could enhance the benign–malignant differentiation of LG tumors. To the best of our knowledge, the role of radiomics in the diffuse pathology of LG including pSS has not been studied so far.

Generally, pSS presents an insidious onset with vague symptoms that cause a frequent delay in diagnosis for many years [22]. Xuan et al. assessed the temporal changes in the exocrine glands in patients with pSS and reported that the parenchymal inflammation first occurs in the LG and is then followed by the major salivary glands’ involvement [23]. Therefore, the aim of our study was to preliminarily assess the value of the textural analysis parameters of the LG on MRI images, which would allow the differentiation of healthy subjects from pSS patients, and to highlight the potential relevance of future radiomic studies in the earlier-imaging diagnosis of pSS.

## 2. Results

A total of 23 patients (mean age 58.7 years old, age range 29–83) diagnosed with pSS were included in this study. The majority of them were female patients (91.3%). The median time between the disease onset and the MRI investigation was 29 months. A total of 86.9% of the subjects presented a positive Schirmer’s test which objectively quantified xerophthalmia. The extended characteristics of the pSS patients are summarized in Table 1. The control group consisted of 23 healthy subjects (mean age 56.3 years old, age range 30–81), 21 of them of the female gender, with no pathological changes detected on cerebral CE-MRI performed for tension-type headaches and migraines (17, 52.2%), cerebral tumor suspicion (5, 21.7%), and ischemic stroke suspicion (6, 26.1%).

A total of nineteen unique texture analysis features showed statistically significant results in the univariate analysis when comparing the pSS group vs. the control group. The Mann–Whitney U test results are displayed in Table 2. These parameters were further included in the multivariate logistic regression analysis, which resulted in a coefficient of determination (R^2^) of 0.48, an adjusted R^2^ of 0.35, and a multiple correlation coefficient of 0.69. Two texture parameters, RunLengthNonUniformityNormalized (*RLNonUN*) and Maximum2DDiameterColumn (*Max2DDC*), were independent predictors for pSS (*p* = 0.04 and *p* = 0.03, respectively) (Table 3). The diagnostic performance of the two predictive radiomic features and the prediction model was assessed by ROC analysis (Table 4 and Figure 2.

The diagnostic performance of the two predictive radiomic features and the prediction model was assessed by ROC analysis (Table 4 and Figure 2). The cutoff value of 5.35 for RLNonUN and 0.77 for Max2DDC differentiates pSS from healthy controls with a sensitivity of 70.83% (CI, 55.9–83%) and 72.92% (CI, 58.2–84.7%), respectively, and a specificity of 70.83% and 79.17%, respectively. The prediction model based on the values expressed by the two independent predictor parameters presented the highest sensitivity (91.67%; CI, 80–97.7%), specificity (83.33%; CI, 69.8–92.5%), and AUC (0.905; CI, 0.828–0.956).

In our group of pSS subjects, no statistically significant correlation was found between the ESSDAI score and the Schirmer’s test values (Table 5).

## 3. Discussion

Our results assess the ability of radiomic features extracted from contrast-enhanced T1-weighted images fat-saturated to differentiate parenchymal changes in LG of patients with pSS from healthy controls.

So far, in the pSS classification criteria, no imaging methods are included, according to international guidelines [6]. However, due to the increased risk of lymphoma development in patients with pSS [4], discovering reliable, non-invasive imaging tools that will allow early diagnosis is crucial. Currently, the SG ultrasound is proposed to be the first-line imaging tool if there is a clinical suspicion of pSS, given its well-recognized diagnostic performance and a validated B-mode OMERACT scoring system [24]. Studies on LG ultrasound (LGUS) in pSS are scarce. One study proved that LGUS is highly reliable in detecting relevant features and is able to distinguish between pSS patients and healthy subjects. Although LG are superficially located and therefore easily accessible with high-frequency ultrasound probes, in some cases, the parenchymal visibility was impaired (up to 11.5% in healthy subjects and 3% in patients with pSS) [12].

The role of MRI in assessing patients with pSS has been widely researched [25], but studies have mainly addressed the structural changes of the major SG. There are few studies that assessed the MRI aspect of LG in pSS, some suggesting that the size change associated with an increased fat deposition [14] and a lower ADC value [13] might be characteristic features of LG affected by pSS. However, the diagnostic performance of these criteria on MRI has not yet been assessed [13,14].

MRI has the advantage of a high soft tissue contrast resolution, providing accurate anatomical details which can be further improved by fat saturation techniques, preventing high signal return from the orbital fat [26]. The current limitations of MRI regarding the LG assessment in pSS might be related to the fact that MRI head–neck protocols are often laborious; they require long times to be performed, and highly specialized radiological personnel is mandatory for correct examinations and radiological reports [27]. No validated eye-related side effects of MRI have been reported in the literature; however, MRI is unsafe to perform on patients with intra-orbital foreign bodies [28,29]. 

As LG are anatomically small-sized glands, diffuse parenchymal changes might be more difficult to detect visually, especially in the early stages. Thus, in this study, we wanted to assess the potential of textural analysis in detecting structural changes of LG in pSS, which are not apparent by simple visual inspection.

Our results show that the Max2DDC parameter extracted from LG parenchyma proved to be an independent predictor for pSS. It held higher values in healthy controls compared to patients with pSS. These might be explained by the progressive degeneration of LG parenchyma caused by the chronic glandular inflammation in pSS, leading to a gradual decrease in the size of LG, even reaching complete atrophy [12].

The RLNonUN parameter was also an independent predictor of affected LG. It displayed higher values for the affected than for unaffected glands. This parameter measures the similarity of run lengths throughout the image, with a lower value indicating more homogeneity among run lengths in the image [30]. This translates into an increased inhomogeneity of LG in patients with pSS. This is in accordance with one preliminary study based on the ultrasound appearance of LG in patients with pSS compared to healthy subjects, where the two most discriminative features were the glandular inhomogeneity and the fibrous aspect of the glands [12,31].

Therefore, it is possible that texture analysis can reflect some of the intrinsic changes of the LG in patients with pSS and can offer adequate differentiation between affected and unaffected glands. However, due to the lack of direct coordination between the MRI examination and a histopathological evaluation, the exact substrate of this differentiation remains unclear and needs to be further evaluated through prospective studies.

The diagnostic performance of the two predictive radiomic features was good, with AUC values of 0.713 and 0.747 for Max2DD and RLNonUN, respectively, while the prediction model using both parameters proved to increase the AUC to 0.905. These results are promising, proving the important role that radiomics has in differentiating between the normal LG structure and the pathological changes that occur in patients with pSS.

To the best of our knowledge, this is the first MRI-based study that assessed the diagnostic value of the radiomic features of LG in patients with pSS.

Radiomic studies on MRI scans have been performed to assess focal lesions of LG but not diffuse pathologies, such as pSS. Lecler et al. also performed a study on subjects with LG lesions and proved that radiomic features extracted from multiple MRI sequences are highly reproducible and generate independent information that might be used as biomarkers [20]. Guo et al. [21] identified four quantitative radiomic parameters, including texture, shape, and intensity features, extracted from MRI T2W imaging and post-contrast T1W imaging that allowed differentiation between benign and malignant lesions of LG with a diagnostic accuracy between 80–86%. Moreover, the resulting combination model presented a superior diagnostic performance (AUC of 0.93), compared to that of radiologists (AUC of 0.70).

The main limitations of this study are the following. Firstly, a small number of patients diagnosed with pSS who also underwent MRI exams were included. This is explained by the relatively low prevalence of this disease in the general population (0.06% worldwide) [32] and the fact that the study was monocentric. However, the monocentric nature of this study allowed all examinations to be performed on the same machine, which resulted in a higher degree of homogeneity of the selected images and, therefore, a more adequate extraction of textural analysis parameters. Both LG of one subject were assessed, leading to an increased number of observations, and a severe Bonferroni correction was applied to counteract any statistical bias. However, future studies on larger cohorts are mandatory to confirm the obtained results and to validate and increase the statistical significance of the assessed correlations. Secondly, the retrospective nature of the study might also have contributed to the selection and verification bias. Thirdly, no biopsy of the LG was performed, as it is not required for the diagnosis of pSS according to the guidelines, this procedure is also more technically challenging and invasive than minor salivary glands biopsy. Nevertheless, it would have allowed the correlation of the imaging aspect with the histopathological changes of LG. In the control group, there was also no histological confirmation that the lacrimal glands were healthy. However, there was neither a clinical or paraclinical aspect nor medical history that would suggest any pathological change in LG in the control group. Finally, we did not assess the parenchymal LG differences between the pSS group and subjects with dry-eye syndrome, which did not fulfill the pSS criteria. This, however, represents an important objective for future research in our department.

## 4. Materials and Methods

### 4.1. Study Groups

This *Health Insurance Portability and Accountability Act*–compliant, single-institution, retrospective pilot study was approved by the institutional review board (ethics committee of the University of Medicine and Pharmacy “Iuliu Hațieganu” Cluj-Napoca; Date of approval: 2 April 2018/No. of approval 166), and informed consent was waived due to the retrospective nature of this research. The study was performed in accordance with the ethical code of the World Medical Association (Declaration of Helsinki).

Between June 2018 and December 2020, 27 patients with previously documented pSS underwent contrast-enhanced MRI examinations of the head–neck region for the assessment of PG and LG. Four patients were excluded from the study due to the severe atrophy of both lacrimal glands, which could therefore not be identified on the MRI scans. Thus, a final number of 23 patients with pSS were included in this retrospective study.

The diagnosis of pSS was established according to the American College of Rheumatology and the European League Against Rheumatism (EULAR) current classification criteria published in 2016 [6]. The clinical examination was performed by one experienced rheumatologist. The severity of the xerophthalmia during the last two weeks was assessed using a 0–2 grading scale questionnaire (0—without symptoms, 1—mild symptoms with symptomatic treatment, 2–severe symptoms even with symptomatic treatment) [7]. The evaluation of xerophthalmia was realized using the Schirmer test in 5 min (mm), and the EULAR Sjögren’s Syndrome disease activity index (ESSDAI) was calculated [33]. After clinical examination, laboratory tests were performed.

The control group included 23 subjects who performed cerebral CE-MRI examination for tension-type headaches and migraines, as well as brain tumor or ischemic stroke suspicion, matched based on age and gender. The control subjects had no pathological changes on the MRI scans and did not present any sicca symptoms, autoimmune disorders, history of head–neck irradiation, or medication that could influence the tears or saliva secretion.

### 4.2. MRI Protocol

The MRI examinations were performed in a single-center, using a 1.5 Tesla MRI scanner (SIGNA™ Explorer, General Electric) with an eight-channel high-resolution head coil. The acquisition protocol consisted of axial T1-weighted imaging fast spin-echo, axial T2-weighted using the Propeller technique (Periodically Rotated Overlapping ParallEL Lines with Enhanced Reconstruction), coronal STIR (Short Tau Inversion Recovery) Propeller, axial diffusion-weighted imaging (DWI) using echo-planar imaging sequences at multiple b-values (b0, b200, b400, b800, and b1000 s/mm^2^) with the corresponding ADC maps; axial perfusion-weighted imaging with enhancement curve generation, coronal gadolinium-enhanced T1-weighted fat-suppressed, and 3D HYDRO T2-weighted imaging.

For the LG textural analysis, the coronal-contrast-enhanced T1-weighted sequence with fat saturation was used, acquired using the following parameters: repetition time/echo time (750/15 ms), field of view (335 × 240 mm^2^), slice thickness (3 mm), slice gap (0.3 mm), and matrix acquisition time (4 min). Post-contrast images were acquired after the intravenous injection of 0.1 mL/kg Gadobudrol (Gadovist; Bayer HealthCare, Berlin, Germany).

### 4.3. Texture Analysis

The radiomics approach consists of four steps: image segmentation using regions of interest, feature extraction, feature selection, and prediction.

#### 4.3.1. Image Pre-Processing and Segmentation

Each examination was reviewed on a dedicated workstation (General Electric, Advantage workstation, 4.7 edition) by one radiology resident (P.A.S.) in his fourth year of training with previous experience in radiomics studies, who reviewed the images for possible artifacts and protocol errors. No examination was excluded for these reasons. All examinations were anonymized, and the selected sequence was retrieved in DICOM format (Digital Imaging and Communications in Medicine) and imported into an open-source texture analysis software, Slicer version 4.11 (available online at: http://www.slicer.org/ (accessed on 1 May 2021). Within the 3D Slicer program, before segmentation, all MR images were preprocessed for intensity normalization and discretization. The 3D segmentation was performed manually by another radiology resident. The researcher incorporated each LG using a three-dimensional region of interest (ROI) using consecutive slices. Both the orbital and the palpebral lobe were included. The segmentations were then independently revised by a senior radiologist with 10 years of experience in head–neck MRI.

An example of LG segmentation is offered in Figure 3.

#### 4.3.2. Feature Extraction

Seven categories of radiomic features were derived from the ROI (region of interest) segmentation of the LG: shape, first-order parameters, GLCM (gray-level co-occurrence matrix), GLDM (gray-level dependence matrix), GLRLM (gray-level run-length matrix), GLSZM (gray-level size zone matrix), and NGTDM (neighboring gray-tone difference matrix). Finally, a total of 103 textural analysis parameters were extracted. The feature extraction was automatically performed by the 3D Slicer software. Previously to the extraction, a ROI normalization was performed to reduce the intensity variations that can affect the true image textures [34].

#### 4.3.3. Feature Selection, Class Prediction, and Statistical Analysis

Two feature-selection steps were used to avoid overfitting in the radiomic model. The absolute values recorded by the two types of fluids for each parameter were compared using a univariate analysis test (Mann–Whitney U). The receiver-operating-characteristic (ROC) analysis was performed, with the calculation of the area under the curve (AUC) with a 95% confidence interval (CI) for the parameters showing *p* values below 0.00052, after the Bonferroni correction (which implied dividing the standard *p*-value of 0.052 to 94; 92 being represented by the number of the extracted features, plus age and gender) on the univariate analysis. A multivariate analysis was conducted to investigate which of the parameters that showed statistically significant results in the univariate analysis were also independent predictors for pSS. This method was used in previous texture analysis studies [17,34] with good classification results. Correlations were assessed using Spearman’s coefficient. Statistical analysis was performed using a commercially available dedicated software, MedCalc version 14.8.1 (MedCalc Software, Mariakerke, Belgium).

## 5. Conclusions

This study proves that radiomics represents an innovative useful method that has the potential to identify reliable textural features of LG that distinguish patients with pSS from controls. Further research into the parenchymal textural changes of LG must be conducted on larger cohorts of patients in order to confirm and validate these results.

## Figures and Tables

**Figure 1 ijms-23-10051-f001:**
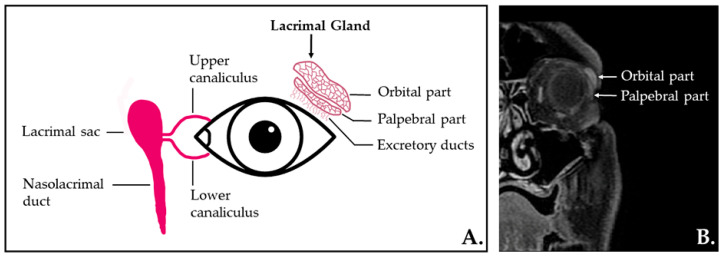
(**A**) The lacrimal apparatus anatomy—illustration. (**B**) Coronal contrast-enhanced T1-weighted MR image with fat saturation depicting the normal aspect of the lacrimal gland localized in the superolateral region of the orbit (extraconal) and its lobe divisions.

**Figure 2 ijms-23-10051-f002:**
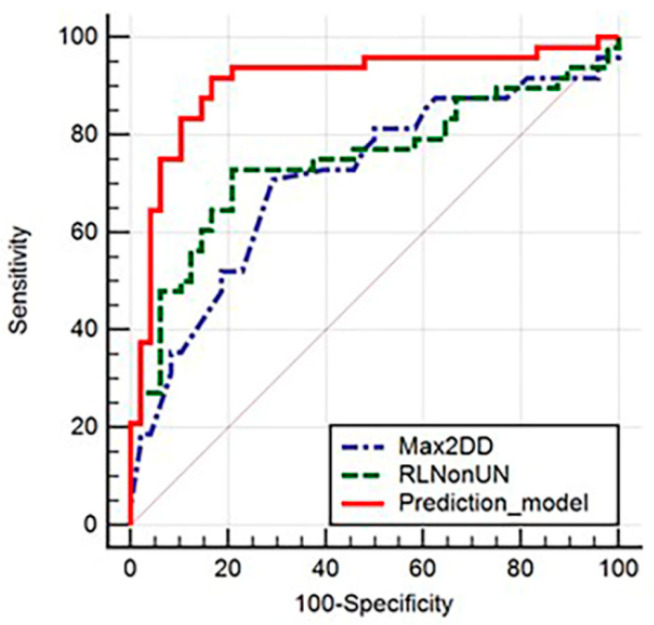
Area under the receiver-operating curve analysis of the two predictive radiomic features (Max2DDC and RLNonUN) and the prediction model for pSS parenchymal changes in LG.

**Figure 3 ijms-23-10051-f003:**
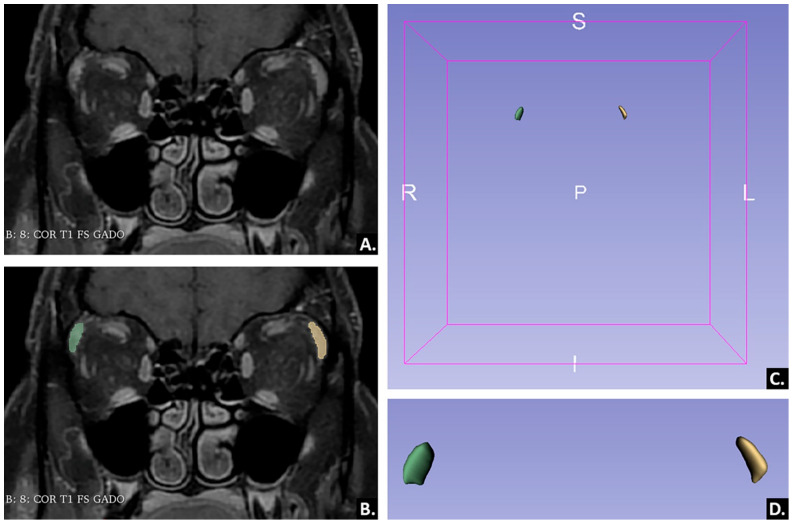
Example of lacrimal gland segmentation in a patient diagnosed with primary Sjögren’s Syndrome. Coronal contrast-enhanced T1-weighed image with fat saturation, before segmentation (**A**) and after segmentation (**B**); 3D reconstruction of the lacrimal glands (**C**); magnified 3D reconstruction of the lacrimal glands (**D**).

**Table 1 ijms-23-10051-t001:** Descriptive data of the pSS patients.

Variable (n = 23)	
Age (years)	58.79 ± 12.64
Female:male	21:2
BMI (kg/m^2^)	26.15 ± 4.78
Disease duration (months)	29 [15.5–60]
ESSDAI	
Inactive	17 (73.9)
Moderately active	4 (17.3)
Severely active	2 (8.69)
ESSDAI score	3.71 ± 5.94
Positive Schirmer’s test	20 (86.9)
Schirmer’s test (mm)	1 [1.5, 3.75]
Xerophthalmia	
Absent	1 (4.3)
Mild under treatment	18 (78.3)
Severe under treatment	4 (17.4)
Anti-Ro/La autoantibodies	20 (86.9)
Rheumatoid factor	18 (78.2)

The results are expressed as mean ± standard deviation, median and [interquartile range], or percentage (%), n = number of patients, BMI = body mass index, ESSDAI = EULAR Sjögren’s Syndrome disease activity index.

**Table 2 ijms-23-10051-t002:** Textural features that show statistically significant results at the univariate analysis between patients with pSS and healthy subjects.

Parameter	*p*-Value	pSS Group (n = 23)		Control Group (n = 23)	
		Median	IQR	Median	IQR
RunPercentage	0.000025	0.89	0.85–0.90	0.85	0.82–0.86
InverseVariance	0.000027	0.30	0.27–0.35	0.36	0.34–0.4
ZoneVariance	0.000028	1.61	0.80–2.44	3.26	2.09–4.27
LargeDependenceEmphasis	0.000029	4.56	3.90–5.96	6.29	5.35–7.92
RunLengthNonUniformityNormalized	0.000031	0.80	0.75–0.84	0.75	0.70–0.77
ShortRunEmphasis	0.000032	0.91	0.89–0.93	0.88	0.86–0.90
GrayLevelNonUniformity	0.000033	8.39	7.04–10.01	11.63	8.88–15.70
SmallDependenceEmphasis	0.000035	0.56	0.47–0.60	0.45	0.41–0.49
DependenceNonUniformityNormalized	0.000047	0.34	0.27–0.37	0.27	0.24–0.30
LongRunEmphasis	0.000050	1.42	1.35–1.57	1.62	1.51–1.82
DependenceVariance	0.000141	1.05	0.79–1.47	1.49	1.28–1.85
SizeZoneNonUniformityNormalized	0.000149	0.53	0.48–0.60	0.47	0.40–0.50
RunVariance	0.000154	0.16	0.12–0.20	0.23	0.18–0.30
SmallAreaEmphasis	0.000189	0.75	0.72–0.80	0.71	0.65–0.74
LargeAreaHighGrayLevelEmphasis	0.000200	1304.26	922.01–2309.91	2835.71	1510.48–4198.58
Kurtosis	0.000335	3.29	2.85–3.84	4.97	3.22–6.73
RobustMeanAbsoluteDeviation	0.000480	71.89	54.54–86.86	55.64	42.45–67.38
Maximum2DDiameterColumn	0.000550	5.03	4.65–5.73	5.77	5.35–6.47
Skewness	0.000550	-0.59	−0.96–−0.11	−1.10	−1.46–−0.44

*p* = statistical significance level; IQR = interquartile range.

**Table 3 ijms-23-10051-t003:** Multivariate analysis results. Bold values are statistically significant.

Independent Variables	Coefficient	Std. Error	t	*p*	r _partial_	r _semipartial_	VIF
DependenceNonUniformityNormalized	−8.77	6.47	−1.35	0.179	−0.15	0.11	141.98
DependenceVariance	−0.60	1.00	−0.60	0.548	−0.06	0.04	220.03
GrayLevelNonUniformity	0.00	0.01	0.48	0.626	0.05	0.04	3.10
InverseVariance	−3.05	1.89	−1.61	0.109	−0.18	0.13	9.69
Kurtosis	−0.02	0.05	−0.39	0.696	−0.04	0.03	6.54
LargeAreaHighGrayLevelEmphasis	−0.00	5.06	−1.03	0.304	−0.11	0.08	5.82
LargeDependenceEmphasis	−1.04	1.31	−0.79	0.431	−0.09	0.06	6242.02
LongRunEmphasis	−5.83	4.41	−1.32	0.189	−0.15	0.10	1440.47
Maximum2DDiameterColumn	−0.09	0.04	−2.12	**0.037**	−0.23	0.17	1.30
RobustMeanAbsoluteDeviation	0.00	0.00	0.93	0.352	0.10	0.07	4.79
RunLengthNonUniformityNormalized	42.15	21.14	1.99	**0.049**	0.22	0.16	1386.23
RunPercentage	−87.08	75.41	−1.15	0.251	−0.13	0.09	8747.97
RunVariance	9.60	7.76	1.23	0.219	0.14	0.10	710.30
ShortRunEmphasis	−82.77	43.88	−1.88	0.063	−0.21	0.15	1750.14
SizeZoneNonUniformityNormalized	−8.85	8.91	−0.99	0.323	−0.11	0.08	473.78
Skewness	0.08	0.12	0.70	0.480	0.08	0.05	4.32
SmallAreaEmphasis	4.35	11.61	0.37	0.709	0.04	0.03	511.95
SmallDependenceEmphasis	21.37	14.73	1.45	0.151	0.16	0.11	1570.01
ZoneVariance	0.06	0.08	0.76	0.449	0.08	0.06	75.30

Std. Error = standard error; *p* = statistical significance level; VIF = Variance Inflation Factor.

**Table 4 ijms-23-10051-t004:** The diagnostic performance of the selected radiomic features and of the combined prediction model in differentiating between normal LG and LG of patients with pSS.

Texture Parameter	AUC	*p*-Value	J	Cut-Off	Se	Sp
Maximum2DDiameterColumn	0.713 (0.611–0.8)	0.0001	0.41	≤5.35	70.83 (55.9–83)	70.83 (55.9–83)
RunLengthNonUniformityNormalized	0.747 (0.648–0.83)	<0.0001	0.52	>0.77	72.92 (58.2–84.7)	79.17 (65.0–89.5)
Prediction model	0.905 (0.828–0.956)	<0.0001	0.75	>0.44	91.67 (80.0–97.7)	83.33 (69.8–92.5)

AUC = area under the curve; J = Youden index; Se = sensitivity; Sp = specificity. Between the brackets are values corresponding to the 95% confidence interval.

**Table 5 ijms-23-10051-t005:** Correlation between the two predictive radiomic features for pSS parenchymal changes in LG and the ESSDAI score and Schirmer test values.

Radiomic Feature	ESSDAI Score	Schirmer’s Test (mm)
Max2DDC	r = 0.315, *p* = 0.134	r = −0.312, *p* = 0.138
RLNonUN	r = −0.191, *p* = 0.372	r = 0.334, *p* = 0.111

r = correlation coefficient; *p* = statistical significance level.

## Data Availability

The data are available only by request.

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
