# Peer review of "The Diagnostic Value of MRI-Based Radiomic Analysis of Lacrimal Glands in Patients with Sjögren’s Syndrome"

_ijms, 2022, doi:10.3390/ijms231710051_

Round 1
Reviewer 1 Report (New Reviewer)
This study investigated the value of lacrimal gland textural analysis parameters on MRI images to see if it could differentiate healthy subjects from pSS patients and also to highlight the potential relevance of future radiomic studies in the earlier imaging diagnosis of pSS.
The study would have been immensely strengthened with the use of another control group, namely subjects that had dry eyes and dry mouth who did not otherwise fulfill the criteria for pSS. That is, a group that had incomplete Sjogren's. The question is how the MRI image would have looked for the LG of subjects with dry eyes, who did not have full blown pSS.
Can the authors correlate the textural changes seen in pSS subjects with salivary gland infiltrates and/or decreased tear flow/eye dryness?
A brief introduction to the criteria for pSS classification needs to be given in the introduction
A cartoon figure of the location of the LG in humans would be beneficial, especially to show how the MRI imaging process will not harm the eye. It would be useful if the authors could discuss this. Also, it would be useful to compare the advantages and disadvantages of using ultrasound versus MRI to image the lacrimal glands. Finally, it would be good to discuss the lacrimal glands in some detail.
Line 105:The Mann-Withney U test?
Author Response
Please see the attachment.

Reviewer 2 Report (New Reviewer)
The authors evaluated the effectiveness of MRI-based texture features of the LG in the patients with primary Sjogren Syndrome (pSS). This is a very unique and interesting study. The comments are described below.
1) Did the authors perform Schirmer test and/or ask the symptom of dry eye to the healthy control group? How many dry eye patients (non-Sjogren type dry eye) were included in the control group? I think that the Schirmer value should be examined in the control group as well.
2) There were extremely few males compared to females in both the pSS group and the control group. I think it would be better to evaluate only women.
3) Did the authors evaluate a correlation between the actual values of the Schirmer test, the ocular symptom score, or the ESSDAI, and the radiomics analysis of LG? If possible, they should add the data. auth may want to add them.
Round 2
Reviewer 1 Report (New Reviewer)
Thanks for addressing this reviewer's concerns. It would have helped if the authors had highlighted the changes made
Reviewer 2 Report (New Reviewer)
The authors responded sincerely and accurately to all comments.
This manuscript is a resubmission of an earlier submission. The following is a list of the peer review reports and author responses from that submission.
Round 1
Reviewer 1 Report
1. Please correct “(RunLengthNonUniformityNormalized, p<0.001 and Maximum2DDiameterColumn” in page 1.
2. Please correct “the aim of our study”. The aim is bold in page 2.
3. Please add the reference of “The severity of xerophthalmia during the last two weeks was appreciated 107 using a 0-2 grading scale questionnaire (0 - without symptoms, 1 - mild symptoms with 108 symptomatic treatment, 2 – severe symptoms even with symptomatic treatment).” in page 3.
4. Please explain what [15.5-60] represent for in table 1.
5. Please make a unified description of p-value in table 2 and p in table 3.
6. Please add the references to these two sentences. “So far, in the pSS classification criteria no imaging methods are included, according 257 to international guidelines.” and “the diagnostic 264 performance of these criteria on MRI has not yet been validated.” in page 7.
7. The authors should explain “Moreover, the LG are anatomically small-sized glands, therefore parenchymal changes might be more difficult to detect, especially in early stages.” more detailed in page 7.
Reviewer 2 Report
In the study there are no data related to symmetry of the presented MRI symptoms in LG. Although pSS is characterized by similar involvement of all exocrine glands and quite similar histopathological pattern in both lacrimal and salivary glands however any differenices/asymetry could be valuable in the assessment of the risk of lymphoma development.
It is postulated to present the results included into the Table 1 separetly for right and left lacrimal glands.
Reviewer 3 Report
The authors precisely stated the purpose of the study, which was to evaluate the effectiveness of MRI-based texture features of lacrimal glands (LGs) in enhancing imaging differentiation between LGs affected by primary Sjögren's syndrome (pSS) and healthy LGs, and to highlight the possible importance of radiomics in the early imaging diagnosis of pSS. Manuscript clearly written. Diagnostically useful results were obtained, i.e., MRI-based texture features can function as quantitative additional criteria that can increase the diagnostic accuracy of LGs affected by pSS.